# Physiologic, Genomic, and Electrochemical Characterization of Two Heterotrophic Marine Sediment Microbes from the *Idiomarina* Genus

**DOI:** 10.3390/microorganisms10061219

**Published:** 2022-06-14

**Authors:** Jorge Vinales, Joshua Sackett, Leah Trutschel, Waleed Amir, Casey Norman, Edmund Leach, Elizabeth Wilbanks, Annette Rowe

**Affiliations:** 1Department of Biological Sciences, University of Cincinnati, Cincinnati, OH 45221, USA; vinaleje@mail.uc.edu (J.V.); sacketja@ucmail.uc.edu (J.S.); trutsclr@mail.uc.edu (L.T.); amirwd@mail.uc.edu (W.A.); normancn@mail.uc.edu (C.N.); leached@ucmail.uc.edu (E.L.); 2Department of Ecology, Evolution and Marine Biology, University of California, Santa Barbara, CA 93106, USA; ewilbanks@ucsb.edu

**Keywords:** electromicrobiology, mineral oxidation, chemolithoheterotrophy, marine, microbiology, sediment microbiology

## Abstract

Extracellular electron transfer (EET), the process that allows microbes to exchange electrons in a redox capacity with solid interfaces such as minerals or electrodes, has been predominantly described in microbes that use iron during respiration. In this work, we characterize the physiology, genome, and electrochemical properties of two obligately heterotrophic marine microbes that were previously isolated from marine sediment cathode enrichments. Phylogenetic analysis of isolate 16S rRNA genes showed two strains, SN11 and FeN1, belonging to the genus *Idiomarina.* Strain SN11 was found to be nearly identical to *I. loihiensis* L2-TR^T^, and strain FeN1 was most closely related to *I. maritima* 908087^T^. Each strain had a relatively small genome (~2.8–2.9 MB). Phenotypic similarities among FeN1, SN11, and the studied strains include being Gram-negative, motile, catalase- and oxidase-positive, and rod-shaped. Physiologically, all strains appeared to exclusively use amino acids as a primary carbon source for growth. This was consistent with genomic observations. Each strain contained 17 to 22 proteins with heme-binding motifs. None of these were predicted to be extracellular, although seven were of unknown localization and lacked functional annotation beyond cytochrome. Despite the lack of homology to known EET pathways, both FeN1 and SN11 were capable of sustained electron uptake over time in an electrochemical system linked to respiration. Given the association of these *Idiomarina* strains with electro-active biofilms in the environment and their lack of autotrophic capabilities, we predict that EET is used exclusively for respiration in these microbes.

## 1. Introduction

Extracellular electron transfer (EET)—the process via which microbes transfer electrons to or from surfaces outside the cell in a redox capacity—is important for both global biogeochemical cycling and biotechnology [1,2,3]. Currently, the best-characterized organisms capable of EET are the metal-oxide-reducing microbes *Shewanella oneidensis* MR-1 and *Geobacter sulfurreducens* [4]. Both of these Gram-negative organisms have similar, but nonhomologous mechanisms of EET. These pathways involve the use of multiple multiheme cytochromes that span from an inner membrane quinol oxidoreductase, across the periplasm, and through to the outer membrane via interactions with an outer membrane porin–multiheme cytochrome complex [4]. Homology to parts of this multiheme cytochrome network, specifically the Mtr system found in *Shewanella*, has been shown in the phototrophic iron oxidizer *Rhodopseudomonas paulustrus* and the neutrophilic iron oxidizer *Siderioxidans lithotrophicus,* providing insight into potential EET mechanisms in these microbes [5,6]. Other nonhomologous but heme-containing proteins involved in iron and/or electrode oxidation in microbes have also recently been identified in the *Zetaproteobacteria Mariprofundus ferroxydans* and the *Deltaproteobacteria* IS5 [7,8,9]. Although reports of EET in a wide range of non-iron-utilizing metabolisms have been highlighted [10,11,12], we have very limited insight into the mechanism and ecology of microbes capable of EET in non-iron-utilizing microbes.

The limited number of mechanisms involved in EET, especially those involved in oxidative-EET (or electron uptake), means that we have limited ability to predict EET from genetic/genomic and/or taxonomic information. To combat this challenge, we used electrochemical techniques in our previous work (electrocultivation) to enrich and isolate microbes capable of electron uptake [13]. Using the biomass from cathodes enriched from Catalina Harbor (CA, USA) marine sediments, we isolated several strains of microbes from eight genera. We described distinct mechanisms of oxidative-EET in the isolated genera on the basis of the mid-point potential of the catalytic activity observed for electron uptake [13]. Each strain was enriched using different techniques, including solid saltwater-based media for targeting sulfur and iron oxidation. Two strains falling within the genus *Idiomarina* were isolated as co-cultivars with different iron- and sulfur-oxidizing strains. Specifically, strains SN11 and FeN1 were isolated from the enrichments that utilized S^0^ or Fe^0^ as an electron acceptor, respectively. These *Idiomarina* strains, however, would only grow axenically on enriched heterotrophic media.

The genus *Idiomarina* was first described in 2000 by Ivanova et. al. [14]. Currently, the genus *Idiomarina* comprises 28 species that have been isolated from seawater and hypersaline environments such as hydrothermal vents [15], fermented fish [16], seashore sand [17], reef-building coral [18], and marine sediment [19]. Despite the habitat diversity among species of *Idiomarina,* all characterized strains are known to grow on amino acids despite being commonly found in mineral-rich and/or halide-rich environments. Other *Idiomarina* strains have also been enriched using electrochemical methods [20]. This work describes the capacity of two *Idiomarina* strains to take up electrons from an electrode in a process that is not linked to growth. Here, we describe the growth characteristics, physiology, genome, and electrochemical activity of these lithoheterotrophic isolates.

## 2. Materials and Methods

### 2.1. Media, Cultivation, and Physiologic Tests

Marine sediment was collected from Catalina Harbor (33°25.23′ N, 118°19.42′ W) in Catalina, CA, USA. Electrochemical incubations from this sediment seeded the first enrichments for microbes capable of electron uptake. Electrodes were subsequently enriched in sediment-free reactors (with nitrate, iron(III)-NTA, and sulfate as electron acceptors), following enrichment with elemental sulfur and iron–sulfur minerals, and subsequent isolation on sulfur or iron(II) plates as previously described [13]. Post isolation, *Idiomarina* cultures FeN1 and SN11 were predominantly maintained using Difco^TM^ Marine Broth (DMB) (Becton, Dickinson and Company, Franklin Lakes, NJ, USA). Glycerol stocks of each strain were maintained in DM medium containing 40% glycerol at −80 °C. Most cultures were grown at room temperature to 30 °C while shaking at 100–150 RPM. Although the post-secondary enrichment used a complex medium (DMB) that allowed for maximum growth, the medium proved difficult to use for growth assays given the high number of suspended particles. Therefore, a particulate-free medium was used with a saltwater base (SWB), containing 20 g of NaCl, 0.113 g of Ca_2_Cl·2H_2_O, 3 g of MgCl_2_·6H_2_O, and 0.5 g of KCl per L and 10 mM NH_4_Cl, 1 mM Na_2_SO_4_, and 1 mM Na_2_HPO_4_, as well as a combination of trace minerals and vitamins [13]. SWB minimal media were used for testing the growth of each strain on various carbon substrates and were used in electrochemical reactors (without added carbon substrates). For more robust growth, another enriched medium was used on the basis of previous work [21], i.e., a Luria–Bertani Broth plus salt and ions (LBS + I) containing 25 g Luria Broth, 10 g of NaCl, 3 g of MgCl_2_·6H_2_O, and 0.1112 g of CaCl_2_·2H_2_O per L. The optimal, temperature, pH, and salinity were tested using the LBS + I medium (minus any NaCl addition in the case of salinity tests). However, for pH growth experiments, additional buffers were added at 10 mM concentrations including a sodium acetate buffer for pH 5.1–6.8 tests, phosphate buffer for neutral pH, Trizma/HCL buffer for pH 7.7–8.1 experiments, and *N*-cyclohexyl-3-aminopropanesulfonic acid for high-pH (pH > 8.1) experiments. The Analytical Profile Index (API) 20NE method (Biomeriuex, France) was used following the manufacturer’s protocol with a 3.5% salt solution on all strains to generally characterize metabolic capabilities. Reference strains from *Idiomarina* (*I. maritima* 908087^T^, *I. donghaiensis* 908033^T^, and *I. loihiensis* L2-TR^T^) were purchased from the ATCC or DSMZ.

### 2.2. Phylogenetic Analysis

Colony PCR was performed using the 8F and 1492R primer sets, which were prepared and used as described previously [21]. Sanger sequencing was performed by a sequencing center (Genewiz, South Plainfield, NJ, USA) in both directions using one of the 8F or 1492R primers. Sequence assembly and QC were performed using the Geneious^TM^ software package (version 2019.2.3), and alignment of ~1400 bp of the 16S rRNA gene to the SIlVA database was achieved using the SINA aligner [22]. Alignment of SN11, FeN1, and nearest neighbors was used to generate a maximum likelihood tree via RAxML (version 8) using the GTR GAMMA Nucleotide model and Rapid Bootstrapping [23]. Gene sequences were deposited to NCBI (Accession numbers: MT162734-MT162735).

### 2.3. Genomic Sequencing and Analysis

Cultures for genomic DNA extraction were grown for 24 h at 30 °C in enriched DMB. For short-read sequencing, the MoBio UltraClean Microbial DNA extraction kit was used to extract and purify DNA samples. Genomic DNA was prepared using Illumina’s Nextera XT sample preparation kit (FC-131-1096) and paired-end sequenced (250 bp reads) using the Illumina HiSeq platform at the UC Davis DNA Technologies Core Facility. DNA for long-read sequencing was extracted using the Qiagen Blood and Tissue Kit (Qiagen, Germantown, MD, USA). Samples were barcoded (Native Barcoding Kit 1D, Oxford Nanopore, UK) and prepared for sequencing (Ligation Sequencing Kit 1D). The library was sequenced using a Flow Cell Mk 1 Spot-ON (FLO-MIN 106 R9). Resulting sequences were base-called with Guppy v. 3.0.6 as implemented in MinKNOW 2.0. Sequence statistics for nanopore reads were calculated with NanoStat v. 1.5.0 [24]. Nanopore long-read sequences were assembled with Flye v. 2.7 [25]. Illumina reads were aligned to the long-read assembly using Burrows–Wheeler aligner v. 0.7.17-r1198 [26] and used to polish the long-read assembly using Pilon v. 1.23 [27] with parameter fix all. Three rounds of polishing were conducted. Genome statistics and assembly quality were determined with QUAST v. 4.4 [28] and CheckM v. 1.0.18 [29].

The genome was annotated using the NCBI Prokaryotic Genome Annotation Pipeline (PGAP) [30]. Potential metabolic pathways were identified using KEGG’s BlastKOALA functional characterization tool [31] available at https://www.kegg.jp/blastkoala/ (accessed on 15 March 2021). The draft and polished genomes were annotated using the NCBI Prokaryotic Genome. Genome sequences can be found under project accession number PRJNA726532. Polished genome accession numbers CP074112 and CP074073 correspond to FeN1 and SN11, respectively (draft genome accession number SN11: WJRO00000000.1, FeN1: WJRP00000000.1). The average nucleotide identity (ANI) was calculated from the genome using an algorithm provided by the EzBioCloud website service (www.ezbiocloud.net; accessed on 1 June 2019) [32]. A digital DNA:DNA hybridization (dDDH) was also performed between SN11 and FeN1 across closely related strains using the genome-to-genome calculator (GGDC) as described by Meier-Kolthoff et al. [33], available on the Leibniz Institute DSMZ website (https://ggdc.dsmz.de/ggdc.php#; accessed on 1 June 2019). The presence of cytochrome-containing genes was investigated by searching for the heme-binding motif (CXXCH) in the predicted open reading frames from each genome using the FIMO (find individual motif occurrences), part of the MEME suite of analysis tools [34]. The program Fe-GENIE was also used to investigate genes putatively involved in iron metabolism [35].

### 2.4. Lipid Analysis

Cultures for fatty-acid analysis were grown overnight in DMB at 30 °C and centrifuged to pellets. The pellets were then resuspended in equal parts DMB and 24% sucrose solution yielding a final 12% sucrose concentration. The suspension was then frozen overnight at −20 °C and placed in a lyophilizer to obtain freeze-dried samples. These samples were sent to the DSMZ for analysis.

### 2.5. Electrochemical Techniques

Cultures for electrochemical experiments were grown aerobically on LBS + I at 30 °C for ~24 h. Cultures were pelleted via centrifugation—7 min, 9000 RPM for FeN1 and 5 min, 9000× *g* for SN11. The pellets were resuspended in SWB lacking NH_4_Cl, Na_2_SO_4_, and Na_2_HPO_4_, as well as minerals, vitamins, and amino acids to limit electrochemically active substrates or potential mediators. The solution was then diluted to an optical density (OD) of 0.4 at 600 nm. The suspended pellets were inoculated into a three-electrode electrochemical system as described previously [13], making the final OD~0.2. This electrode system comprises an indium tin oxide (ITO)-coated glass working electrode, an Ag/AgCl reference electrode, and a Pt wire counter electrode submerged in 20 mL total volume. Each reactor was purged with oxygen during experiments, and a potentiostat (Biologic, Seyssinet-Pariset, France) was used to bias the working electrode to −200 mV vs. SHE. Cyclic voltammetry was run using a 1 mV/s scan rate (unless otherwise specified) from +200 mV to −800 mV. Turnover CVs were run with room air bubbled through the reactors, while non-turnover CVs were run with reactors purged with high-purity nitrogen for at least 1 h prior to running the CV. The strains were tested in at least triplicate, and an abiotic control containing only 20 mL of SWB-L was used to measure a baseline current. Sterile, deionized water was used to replace evaporated media. Additional mediators were also tested, including FMN (2 µM) and L-cysteine (100 µM), which were added from filter-sterilized stock solutions. Antimycin A was added from a 200 mM stock solution in DMSO to a final concentration of 100 µM, to quantify the biological contribution of current as described previously [36].

### 2.6. Scanning Electron Microscopy

Biofilms were imaged on the ITO electrodes (post chronoamperometry) that were fixed with 2% glutaraldehyde in SWB for 24 h at 4 °C. The fixed samples were then dehydrated via a series of ethanol washes of increasing concentration followed by a final dehydration step using hexamethyldisilazane as described in [37]. Samples were then sputter-coated with gold. Scanning electron microscopy was performed using a ThermoFisher Scientific Apreo SEM (University of Cincinnati Advanced Materials Characterization Center).

### 2.7. Fluorescence Microscopy

Live-cell imaging was performed on strain SN11 using previously described techniques [38]. In brief, cells were imaged across an ITO-plated glass coverslip (SPI supplies, West Chester, PA, USA) using a Nikon TI-E inverted microscope equipped with UV fluorescence detection. Cells were visualized using the lipid stain FMTM 4-64FX (Molecular Probes, Life Technologies, Carlsbad, CA, USA) at concentrations specified by the manufacturer. RedoxSensor^TM^ Green (RSG) (Molecular Probes, Life Technologies) studies were performed by adding approximately a 10 µM concentration of RSG in addition to lipid stain. Images were taken at a fixed point on the coverslip at 5 min intervals. At the beginning of the experiment, the cathode potential was poised at −200 mV vs. SHE and then increased to −400 mV vs. SHE (occurs at 9 s in Appendix A).

## 3. Results and Discussion

### 3.1. Lithoheterotrophic Isolates Fall within the Genus Idiomarina

Phylogenetic analysis of the FeN1 and SN11 strains was performed using 16S rRNA gene sequencing of an isolated colony (Figure 1). Strain SN11 displayed a 16S rRNA sequence similarity of 99.77% with type strain *I. loihiensis* L2TR^T^. Strain FeN1 shared a 16S rRNA sequence similarity of 98.21% with type strain *I. maritima* 908087^T^. Another closely related strain, *I. donghaiensis* 908033^T^, shared about 97.36% and 94.87% 16S rRNA identity with FeN1 and SN11, respectively, and was also included in further phylogenetic analyses.

To further evaluate phylogeny, genomic features were compared across the closely related *Idiomarina* strains. The ANI values between FeN1 and related strains *I. maritima* 908087^T^, *I. donghaiensis* 908033^T^, and *I. loihiensis* L2-TR^T^ were 73.49%, 73.36%, and 69.63%, respectively (Appendix B, Table A1). The ANI values between SN11 and related strains *I. maritima* 908087^T^, *I. donghaiensis* 908033^T^, and *I. loihiensis* L2-TR^T^ were 69.62%, 69.80%, and 97.46%, respectively (Appendix B, Table A1). FeN1 displayed a very different genome in comparison to its closely related strains. However, SN11 did not meet the standard ANI criteria for species identity of 95–96% when compared to *I. loihiensis* L2-TR^T^, confirming that this organism is a new strain of the *Idiomarina loihiensis* species. Notably, *I. loihiensis* L2-TR^T^ was isolated from newly formed marine sediments at Lō’ihi seamount [15], while SN11 was isolated for iron-rich sediments in Catalina harbor—sites that are separated by over 3500 km.

The dDDH values, calculated by the GGDC, between FeN1 and SN11, *I. maritima* 908087^T^, *I. donghaiensis* 908033^T^, and *I. loihiensis* L2-TR^T^ were 23.0%, 19.1%, 20.1%, and 24.3%, respectively (Appendix B, Table A1). These values are below the standard (70%) for the identification of a new species, suggesting that FeN1 is a new species. The dDDH values between SN11 and *I. maritima* 908087^T^, *I. donghaiensis* 908033^T^, and *I. loihiensis* L2-TR^T^ were 23.2%, 23.5%, and 75.7%, which again supports SN11 as a new strain of *I. loihiensis* (Appendix B, Table A1). The G + C content of the chromosomal DNA was obtained using the Integrated Microbial Genomes and Microbiomes (IMG) analysis service provided by the Joint Genome Institute (jgi.doe.gov/). All values were similar to related *Idiomarina* strains: 48.15% for FeN1, 47.02% for SN11, 48.36% for *I. donghaiensis* 908033^T^, 47.22% for *I. maritima* 908087^T^, and 47.04% for *I. loihiensis* L2-TR^T^ (Table 1).

On the basis of phylogenetic similarity, the type strains *I. maritima* 908087^T^ (EU600203), *I. donghaiensis* 908033^T^ (EU600204), and *I. loihiensis* L2TR^T^ (AF288370) were also used for reference in comparative lipid analysis. An overall similar lipid composition was observed across the *Idiomarina* strains tested. The predominant lipids observed among all strains were C_15:0_ iso, C_17:0_ iso, C_17:1_ iso, and C_16:0_ (Table 2). Strain FeN1 showed a lower representation of C_15:0_ iso lipids (10–20% difference), compared with all the closely related strains. Notably, this was the most abundant fatty acid among all strains. FeN1 was moderately enriched in C_17:0_ iso lipids compared with other strains. Both strains isolated from Catalina cathode incubations, SN11 and FeN1, were enriched in C_16:0_ lipids compared to other tested strains. This work supports chemo-physiological differences between these closely related strains.

### 3.2. Growth of Idiomarina Occurs Exclusively in the Presence of 11 Amino Acids

An initial screen for metabolic activity in FeN1 and SN11 was performed using the Analytical Profile Index (API) 20NE method. The majority of tests for strains SN11 and FeN1, as well as the reference strains, showed negative results. Negative results were confirmed for the assimilation of all organic compounds tested on the API 20NE strips: glucose, arabinose, mannose, mannitol, *N*-acetyl-glucosamine, maltose, potassium gluconate, capric acid, adipic acid, malate, trisodium citrate, and phenylacetic acid. Growth on additional carbohydrates was tested in SWB containing 10 mM concentrations of respective carbon sources: acetate, citrate, lactate, and fumarate. Strains SN11 and FeN1, along with reference strains, showed negative growth on these carbon sources. The existing literature further suggests that *Idiomarina* can grow via amino-acid metabolism as opposed to using sugars, likely due to the loss of sugar transport systems [40]. Consequently, we tested the growth of these strains on minimal medium containing various amino acids. Minimal SWB medium was amended by adding 20 mg/L of different amino acids to test requirements for the growth of all strains. No single amino acid could support growth of either strain; thus, combinations of various amino acids were tested. Growth for all strains could be supported in media containing the following amino acids: lysine, phenylalanine, tyrosine, tryptophan, histidine, arginine, isoleucine, leucine, serine, threonine, and valine. Addition of carbon sources (i.e., acetate, citrate, lactate, and fumarate) in combination with the amino acids did not promote enhanced growth relative to the experiments with amino acid only. Amino acids were used for growth in all subsequent work with SWB medium, but were omitted for tests in electrochemical systems and for growth tests on other carbon sources.

Using enriched medium (LBS + I), we compared the temperature, pH, and salinity tolerances for each strain. A similar temperature range for optimal growth was expected for both SN11 and FeN1 given their presence in the same geographical region. However, SN11 showed optimum growth at 25–30 °C, while FeN1 preferred temperatures ranging from 35 to 40 °C (Table 1). SN11 showed minimal growth at temperatures closer to 40 °C, which is consistent with the growth range of type strain *I. loihiensis* L2TR^T^. Despite this difference in preferred temperatures, both SN11 and FeN1 showed an identical range of salinity content for growth. Both strains grew in a range of 0.1–18% NaCl (*w/v*, %) (Table 1). This places SN11 and FeN1 under the category of moderate halophiles and suggests diversity in physiological adaptations to varying environmental factors. *I. loihiensis* L2TR^T^ displayed a similar range for growth of 0.25–17.5% NaCl; however, *I. donghaiensis* 908033^T^ seemed to be less resistant to increases in salinity given its growth range of 0.1–12.5% salt, while *I. maritima* 908087^T^ displayed growth in the absence of NaCl with a growth range of 0–13%. Growth in varying pH environments was tested from pH 5.15 to 10.2 (Table 1). FeN1 displayed growth from pH 6.1–10, and SN11 grew at pH 6.1–9.4. All strains displayed similar pH growth ranges varying from a pH of ~6 to 10.

Presently, oxygen is the only known electron acceptor for these organisms. Both strains SN11 and FeN1 were positive for catalase and cytochrome oxidase (Table 1). Notably, SN11 and FeN1 contained three and five annotated catalase or peroxidase enzymes in their genomes, respectively. The API 20NE strips showed negative enzymatic results for both SN11 and FeN1 for nitrate reduction. This was also supported by genomic observations, with each strain lacking a complete denitrification pathway. Each strain encoded only a nitrite reductase (NO-forming) and nitric oxide reductase. Negative enzyme reactivity results were also observed for indole production, fermentation of glucose, arginine dihydrolase, urease, hydrolysis of beta-glucosidase, and beta-galactosidase. A positive test result was observed for the hydrolysis of gelatin in FeN1, *I. loihiensis* L2TR^T^, and *I. donghaiensis*, while SN11 and *I. maritima* displayed a negative result for this test (Table 1). These results further elucidate the differences in metabolism between SN11 and *I. loihiensis*, despite belonging to the same species [21].

### 3.3. Streamlined Genome Supports a Narrow Environmental Niche for Growth

A complete assembly of the genomes of FeN1 and SN11 was obtained from a combination of Illumina and Nanopore MinIon sequencing. Genomic assembly resulted in relatively small genomes for these Gammaproteobacteria, with genomes of 2.9 to 2.8 MB, respectively (Table 3). Each genome contained approximately 2600 protein-coding genes, 56–57 tRNAs, and four rRNA operons (Table 3), suggesting a similar transcriptional, translational, and growth ecology. Given that each genome had a completeness of >99%, we investigated evidence for metabolic and ecologic strategies in these strains.

Each *Idiomarina* strain contained complete glycolysis and TCA cycles. As these organisms were incapable of growth on carbohydrates including glucose, it is likely that these pathways serve as a central metabolism that allows amino-acid metabolism to fuel both biosynthesis and respiration; alternatively, the metabolism of glucose is limited by features such as transport. Given their dependency on amino acids for growth, it is not surprising that these strains were auxotrophic for several amino acids including arginine, isoleucine, phenylalanine, histidine, leucine, threonine, valine, proline, and tyrosine. Growth without amendment of these amino acids (with the exception of omitting proline) was not observed. Although not genomically auxotrophic, growth was dependent on the addition of tryptophan, serine, and lysine, perhaps indicative of a nonfunctional pathway in the genome or environmental availability.

The presence of peptidases in the *Idiomarina* genomes was also investigated, assuming these enzymes could serve as a source or amino acids in the environment. Peptidases are enzymes capable of cleaving peptide bonds in proteins and peptides and play important physiologic roles [41]. In the SN11 strain, the most abundant peptidase was of subfamily M23B, beta-lytic metallopeptidase. We 24 identified M23B peptidases in the SN11 genome, which function to lyse cell walls of other bacteria as either a defensive or a feeding mechanism. The second most abundant peptidase in the same genome was the S33 peptidase, prolyl aminopeptidase, which allows a selective advantage toward proline-rich substrates due to signal peptides that are secreted or periplasmic enzymes. This highlights a potential environmental source for proline, for which these organisms are putative auxotrophs.

Each *Idiomarina* strain contained a complete aerobic electron transport chain for oxidative phosphorylation. Genes involved in the reduction of alternate terminal electron acceptors were sporadic. The genome of FeN1 encoded a sulfide:quinone oxidoreductase sqr, which catalyzes the oxidation of hydrogen sulfide to polysulfide while donating two electrons to the quinone pool. Both strains contained incomplete denitrification pathways. The genome lacked homologs to genes known to be involved in EET, iron reduction, and iron oxidation. FeGenie identified five and seven genes invovld in iron regulation in FeN1 and SN11, respectively, and three and four genes involved in iron storage, respectively. Consequently, we investigated the presence of heme-binding proteins (containing a CXXCH motif). SN11 contained 17 putative heme-binding proteins while FeN1 contained 28 (Table 4). Between the two strains, 13 cytochromes were shared, and six of these lacked a significant functional annotation. They all contained a periplasmic, membrane, or unknown localization signal which supports a possible affiliation with an EET pathway, which will be investigated further.

### 3.4. Electrochemical Activity Suggests That Electron Uptake Is Involved in Respiration (Not Growth)

Electrochemical characterization experiments were carried out to confirm the ability of the *Idiomarina* strains to perform extracellular electron uptake. Using chronoamperometry, which measures current over time at a controlled redox potential, strains FeN1 and SN11 were tested for electrochemical properties. Chronoamperometry experiments showed that both SN11 and FeN1 sustained cathodic current production (negative currents) in the ITO-plated bioreactors compared to the abiotic control (Figure 2). Strain FeN1 showed a consistent current production of −0.5 µA, whereas SN11 sustained a current of −0.8–0.9 µA (negative currents indicative of electron uptake). Antimycin A, an inhibitor of cellular respiration through inhibition of cellular ubiquinone and cytochrome c interactions, was used to demonstrate the biological nature of cathodic current production. Current in biological reactors decreased upon the addition of Antimycin A (Figure 2). The abiotic control was not affected by the addition of Antimycin A and showed only background current production (~0.01 µA) in the duration of the experiments (Figure 2). Cyclic voltammetry demonstrated the relationship between current and voltage for each strain and demonstrated a distinct increase in cathodic (negative) current around −100 mV vs. SHE (Figure 2B,C). Non-turnover CVs supported subtle differences in the strains (Figure 2D,E). Specifically, the mid-point potentials measured for redox active features corresponding to the turnover electron uptake feature varied from −106.5 ± 7 mV vs. SHE for SN11 to −122 ± 12 mV vs. SHE for FeN1.

SEM images indicated that both SN11 and FeN1 were 1 to <2µm long rods approximately 0.25 μm in width (Figure 3), which is consistent with the compared strains (Table 1). Notably, SN11 showed small, ubiquitous protrusions along its outer membrane. Noteworthily, cell attachment to the electrodes observed via SEM was poor. Very few cells, separated by larges distances, were observed. Consequently, we used live imaging to observe cells in vivo in an electrochemical system (Appendix A). Clumped groups of cells can be seen loosely attached to the electrode and moving in and out of focus. Cells were stained with a lipid stain and redox sensor green, which has been used in *Shewanella* to indicate electron flow from a cathode to high-potential cytochromes [38]. Given the challenges of consistently visualizing cells over time in this system, we were unable to quantify RSG fluorescence with the various conditions, although we qualitatively noted an increase in RSG signal when the electrode was poised from −200 to −400 mV vs. SHE (at ~9 s in Appendix A). Loose attachment of cells to the electrode is also consistent with the observation that removing the medium from electrochemical reactors dramatically reduced biological activity in the electrochemical system Although cells did not appear to attach well to surfaces, they did for large cells clumps or aggregates >100 cells. Staining of the aggregates with the nucleic acid stain SYBR green showed stained filaments. This was potentially DNA, as it has been shown to be incorporated into biofilms of other microbes and could make up a significant component of these aggregates (Appendix B; Figure A1).

Because of their loose attachment to electrodes, we investigated the potential for redox active mediators to play a role in *Idiomarina* electron transfer. Spent media from electrochemical reactors was centrifuged to remove cells and added to a clean electrochemical system. No evidence of redox active mediators was observed; however, given the nature of the cell aggregates recovered, it is possible that mediators were concentrated in or around the aggregates, or they co-localized and interacted with the DNA in biofilms, as has been observed in phenazine-producing *Pseudamonas* cultures [42]; hence, they would have been removed with centrifugation. As such, we added traditional redox active compounds that have been shown to serve as mediators in other electrochemical active microbes, specifically flavins (i.e., FMN) and the amino acid cysteine [43,44,45]. Cysteine addition resulted in a significant overall drop in current for FeN1 cultures and limited to no drop in SN11 and control reactors (Appendix B; Figure A2). We predict that this was likely due to a switch in metabolic strategy from electron uptake to amino-acid consumption. Conversely, addition of FMN to each reactor did increase the overall current in both biological and abiotic control reactors, while no difference was noted in the range of biological current being consumed in these experiments, as shown through the addition of antimycin (Appendix B; Appendix B). Specifically, all biologic currents fell within the 2 to 0.5 µA range observed for both FeN1 and SN11. Consequently, we predict that FMN addition increased the background/abiotic current levels rather than biologic activity. As such, there is currently no evidence for a mediated electron transfer interaction between cells and the electrode in these strains.

## 4. Conclusions

This study outlined the electrochemical activity of two microbes that obligately grow using amino acids as a substrate. Given these results, we believe that SN11 and FeN1 are promising candidates for further exploring electron transfer among bacterial microorganisms at a physiological and genetic level, in part due to their capacity for lithoheterotrophy. Although these microbes are only “weak” electrotrophs, they consistently perform electron uptake that is uncoupled from their growth physiology. This is similar to organisms such as *Shewanella* that can perform electron uptake from a cathode uncoupled to growth [38]. However, unlike *Shewanella*, there is no precedence for EET in the *Idiomarina* genus. This organism provides new insight into a different ecological role of EET in marine microbes, specifically looking at one that is not linked to mineral metabolism. Although it is not yet clear what role EET plays in *Idiomarina* physiology and ecology, given that this microbe is often isolated from iron- and sulfur-oxidizing environments (and does not itself perform these metabolisms) [14,15,40], we speculate that EET may be used to help *Idiomarina* obtain energy in the presence of a conductive biofilm, potentially in a parasitic nature when amino acids are limited.

## Figures and Tables

**Figure 1 microorganisms-10-01219-f001:**
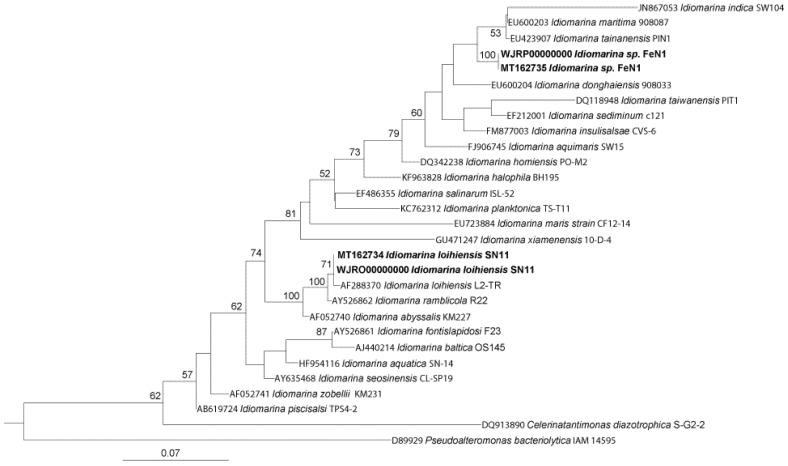
Phylogenetic placement of SN11 and FeN1 (shown in bold) based on approximately 1400 bp of the 16S rRNA gene aligned using SINA aligner [13]. The maximum likelihood phylogenetic tree was generated via RAxML version 8 [17]. Bootstrap values > 50% for 100 trees are shown for each node. The scale bar indicates the mean number of nucleotide substitutions per respective branch.

**Figure 2 microorganisms-10-01219-f002:**
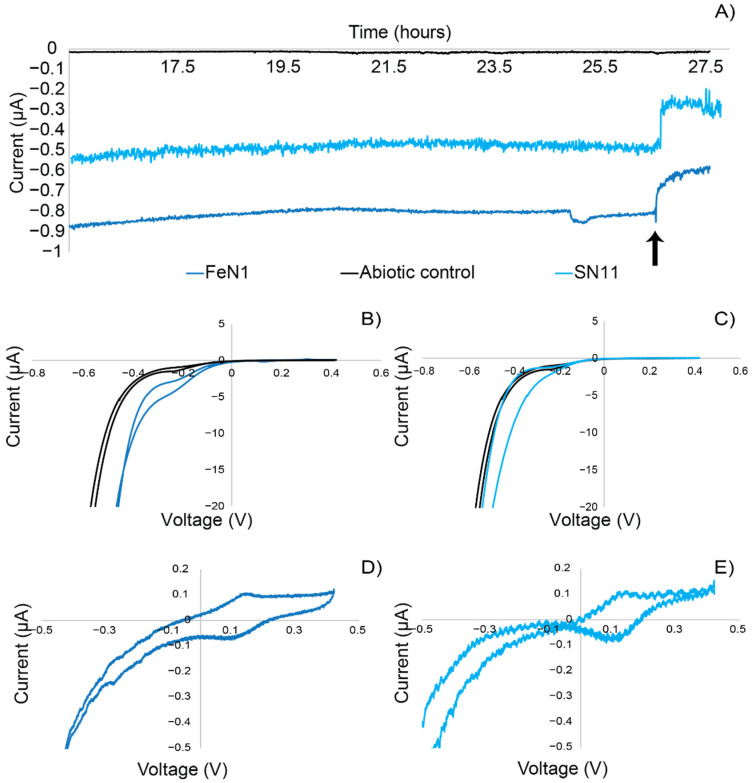
Example plots of (**A**) *Idiomarina* chronoamperometry for SN11 and FeN1 vs. an abiotic control. The current was measured for electron uptake at −202 mV vs. SHE through antimycin addition (at 26 h, indicated by black arrow), which resulted in a decreased cathodic current in FeN1 and SN11. The current was normalized to electrode area of flat ITO-plated glass electrodes. (**B**,**C**) Turnover CVs of FeN1 and SN11 respectively, run in the presence of O_2_. (**D**,**E**) Non-turnover CVs of FeN1 and SN11 respectively, run while purging with N_2_. CVs were run at 1 mV/s.

**Figure 3 microorganisms-10-01219-f003:**
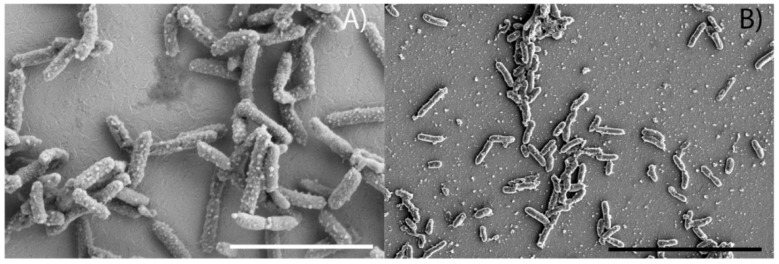
Scanning electron microscopy images of glutaraldehyde-fixed, ITO-plated electrodes. Electrodes taken from SN11 (**A**) and FeN1 (**B**) bioreactors post chronoamperometry. The scale bar in panel (**A**) is 4 µm and the scale bar in panel (**B**) is 10 µm.

**Table 1 microorganisms-10-01219-t001:** Phenotypes displayed by FeN1 and related *Idiomarina* strains. 1, FeN1; 2, SN11; 3, *I. loihiensis* L2TR^T^; 4, *I. donghaiensis* 908033^T^; 5, *I. maritima* 908087^T^.

Characteristics	1	2	3	4	5
Cell length (µm)	1.0–2.0	1.0–2.0	0.7–1.8 ^1^	1.0–1.4 ^2^	1.4–2.0 ^2^
Temperature ranges (°C)	4–40	4–40	6–39	7–42.5	7–45
Optimum temperature (°C)	35–40	25–30	nt	nt	nt
NaCl ranges (%, *w/v*)	0.1–18	0.1–18	0.25–17.5	0.1–12.5	0–13
pH ranges	6.1–10	6.1–9.4	6.5–9.4	6.1–10	6.5–9.4
G + C content (mol.%)	48.15	47.02	47.04	48.36	47.22
Metabolic Tests					
Catalase	+	+	+	+	+
Oxidase	+	+	+	+	+
Hydrolysis of protein (gelatin)	+	−	+	+	−

^1^ data taken from [15], ^2^ data taken from [39]. nt stands for “not tested”.

**Table 2 microorganisms-10-01219-t002:** Percentage fatty-acid composition. 1, FeN1; 2, SN11; 3, *I. loihiensis* L2TR^2^; 4, *I. donghaiensis* 908033^T^; 5, *I. maritima* 908087^T^*. The* range of error for lipid measurements varied by ±0.005–0.300 between replicate injections.

Fatty Acid	1	2	3	4	5
C_11:0_ iso	1.9	2.3	2.2	2.44	1.9
C_11:0_ iso 3OH	4.1	4.1	4.5	4.26	4.7
c_13:0_ iso	1.6	2.2	1.6	3.38	2.1
c_13:0_ iso 3OH	3.8	3.8	3.6	3.39	3.7
c_15:1_ iso F	1.8	3.2	1.6	5.42	5.9
c_15:0_ iso	26.0	38.4	30.4	36.82	32.8
c_16:0_	7.4	7.2	1.1	2.88	3.8
c_17:0_	2.0	1.5	1.3	0.35	5.2
c_17:0_ iso	15.0	11.8	17.1	13.2	14.2
c_17:1_ iso	11.5	12.0	11.6	9.53	14.0
c_18:0_	2.7	1.1	1.8	2.22	2.0

**Table 3 microorganisms-10-01219-t003:** Assembly and quality statistics for *Idiomarina* sp. FeN1 and SN11 genomes.

	FeN1	SN11
BioProject accession no.	PRJNA726532	PRJNA726532
Assembly accession no.	CP074112	CP074073
Assembly size (bp)	2,900,277	2,834,848
Estimated genome completeness (%) ^1^	99.66%	100%
Estimated contamination (%) ^1^	0.9%	0.17%
No. of contigs	1	1
No. of protein-coding genes	2667	2616
No. of tRNA genes	57	56
No. of rRNA operons	4	4

^1^ Determined with CheckM v1.0.18 [29].

**Table 4 microorganisms-10-01219-t004:** Predicted localization, based on PSORTb algorithm, of proteins containing putative CXXCH heme-binding motifs in each *Idiomarina* genome.

SN11 Locus Tag	FeN1 Locus Tag	Annotation	Query Coverage/Percentage Identity ^1^	Localization (Score)
KF946_02970	KGF88_02385	Cytochrome c	98/54	Periplasmic (10.00)
KF946_04255	KGF88_00295	NAD(P)-dependent alcohol dehydrogenase	99/61	Cytoplasmic (9.97)
KF946_04965	KGF88_04300	Cytochrome-c oxidase, cbb3-type subunit III ccoP	99/74	Unknown
KF946_04975	KGF88_04310	Cytochrome-c oxidase, cbb3-type subunit II ccoO	97/85	Cytoplasmic (8.96)
KF946_05620		4a-Hydroxytetrahydrobiopterin dehydratase		Cytoplasmic (9.26)
KF946_05805	KGF88_01080	Cytochrome c	94/60	Unknown/periplasmic (9.84)
KF946_05965		DUF3179 domain-containing protein		Cytoplasmic membrane (10.00)
KF946_06450	KGF88_06085	2Fe–2S iron–sulfur cluster binding domain-containing protein	100/67	Unknown
KF946_06970	KGF88_07065	Molecular chaperone DnaJ	99/79	Cytoplasmic (9.97)
KF946_09505	KGF88_09595	Cytochrome c1	100/73	Unknown—may have multiple localization sites
KF946_10365	KGF88_10235	Cytochrome c oxidase subunit II coxB	98/74	Cytoplasmic membrane (9.99)
KF946_10655	KGF88_10495	Cytochrome c5 family protein	88/59	Unknown
KF946_10745	KGF88_12320	Anaerobic ribonucleoside-triphosphate reductase activating protein nrdG	92/72	Cytoplasmic (9.26)
KF946_11095		Cytochrome c/FTR1 family iron permease		Cytoplasmic membrane (10.00)
KF946_11525	KGF88_11325	Cytochrome c4	100/64	Periplasmic (10.00)
KF946_12560	KGF88_12535	50S ribosomal protein L31 rpmE	97/70	Cytoplasmic (9.26)
KF946_13430		EAL and GGDEF domain-containing protein		Cytoplasmic membrane (7.88)
	KGF88_01265	dsbC family protein		Periplasmic (9.76)
	KGF88_01425	Thioredoxin family protein		Unknown—may have multiple localization sites
	KGF88_01810	Cryptochrome/photolyase family protein		Unknown
	KGF88_05965	Thioredoxin trxC		Cytoplasmic (9.26)
	KGF88_07240	Flp pilus assembly complex ATPase component TadA		Cytoplasmic (9.97)
	KGF88_09010	ISC system 2Fe-2S type ferredoxin		Cytoplasmic (8.96)
	KGF88_09905	Exinuclease ABC subunit UvrA		Cytoplasmic (9.97)
	KGF88_10910	Catalase		Periplasmic (10.00)
	KGF88_11150	Cytochrome c peroxidase		Periplasmic (10.00)
	KGF88_11220	DUF3365 domain-containing protein		Periplasmic (9.84)
	KGF88_13155	C-type cytochrome		Cytoplasmic membrane (10.00)
	KGF88_13170	C-type cytochrome		Periplasmic (10.00)
	KGF88_13175	C-type cytochrome		Periplasmic (10.00)
	KGF88_13495	Rhodanese-related sulfurtransferase		Cytoplasmic (8.96)
	KGF88_13780	C-type cytochrome		Periplasmic (10.00)

^1^ Query coverage and percentage identity calculated using reciprocal protein–protein BLAST searches.

## Data Availability

Sequences have been deposited and made available in NCBI. Ribosomal sequences can be under accession numbers: MT162734–MT162735. Genome sequences can be found under project accession number PRJNA726532. Polished genome sequences can be found under accession numbers CP074112 and CP074073.

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
