# Peer review of "Physiologic, Genomic, and Electrochemical Characterization of Two Heterotrophic Marine Sediment Microbes from the Idiomarina Genus"

_microorganisms, 2022, doi:10.3390/microorganisms10061219_

Round 1

Reviewer 1 Report

Review of ‘Physiologic, Genomic and Electrochemical Characterization of two heterotrophic marine sediment microbes from the Idiomarina genus.

The authors studied physiology, genome, and electrochemical properties of two marine microbes belonging to the genus Idiomarina from marine sediment cathode enrichments. The authors defined these organisms as I. loihiensis and I. maritime. The authors concluded that the microbes use amino acids as a primary carbon source and extracellular electron transfer is used for respiration only. This study is well-designed. The paper is well-written and illustrated. The new data expand our knowledge of the Idiomarina physiology in the marine environment.

Abstract.

Pg 1 Ln 10: Suggest changing ‘Extracellular electron transfer’ to ‘Extracellular electron transfer (EET)’

Pg 1 Ln 11: Suggest changing ‘mineral’ to ‘minerals’

Introduction

Appropriate.

Methods

I suggest including some information about the study site (location, depth, substrate, etc) from which these species were isolated.

Pg 2 Ln 89: Suggest changing ‘was tested’ to ‘were tested’

Pg 3 Ln 129: Suggest changing ‘sequences found’ to ‘sequences were found’

Results and discussion.

Pg 4 Ln 190: The authors should italicize ‘I. loihiensis‘

Pg 4 Ln 191: The authors should italicize ‘I. maritima‘

Pg 4 Ln 192: The authors should italicize ‘I. donghaiensis‘

Latin names should be italicized throughout the text.

Pg 5 Ln 217: Suggest changing ‘new strain’ to ‘a new strain’

Pg 7 Ln 253: Suggest changing ‘medial’ to ‘media’

Pg 9 Ln 318: Suggest changing ‘which allow’ to ‘which allows’

Pg 9 Ln 322: Suggest changing ‘Idiomarina strains’ to ‘Idiomarina strain’

There are only a few citations in this section. The authors should include more references and expand the discussion.

Conclusions.

Pg 12 Ln 410: Suggest changing ‘its growth physiology’ to ‘their growth physiology’

Author Response

We thank the reviewer for their feedback and we have made the following changes based on their suggestions. All responses in bold/italics.

Abstract.

Pg 1 Ln 10: Suggest changing ‘Extracellular electron transfer’ to ‘Extracellular electron transfer (EET)’

This has been changed

Pg 1 Ln 11: Suggest changing ‘mineral’ to ‘minerals’

This has been changed

Introduction

Appropriate.

Methods

I suggest including some information about the study site (location, depth, substrate, etc) from which these species were isolated.

This has been added to the text: line 77-82 (page 2)

Pg 2 Ln 89: Suggest changing ‘was tested’ to ‘were tested’

Use of was has been changed to were in this section.

Pg 3 Ln 129: Suggest changing ‘sequences found’ to ‘sequences were found’

This has been changed

Results and discussion.

Pg 4 Ln 190: The authors should italicize ‘I. loihiensis‘

This has been changed

Pg 4 Ln 191: The authors should italicize ‘I. maritima‘

This has been changed

Pg 4 Ln 192: The authors should italicize ‘I. donghaiensis‘

This has been changed

Latin names should be italicized throughout the text.

We apologize for these typo's. This is likely due to copying into the MDPI format which did not preserve all the formatting. Sorry we did not catch these, but the manuscript has been updated accordingly.

Pg 5 Ln 217: Suggest changing ‘new strain’ to ‘a new strain’

This has been changed

Pg 7 Ln 253: Suggest changing ‘medial’ to ‘media’

This has been changed

Pg 9 Ln 318: Suggest changing ‘which allow’ to ‘which allows’

This has been changed

Pg 9 Ln 322: Suggest changing ‘Idiomarina strains’ to ‘Idiomarina strain’

This has been changed

There are only a few citations in this section. The authors should include more references and expand the discussion.

As requested by the other reviewer we have added some additional analysis of iron metabolism, and EET. Additional references and discussion has been added to the text (Line 146-150 & 346-353). 

Additional discussion and citations on peptidases have been added (Line 329-339).

Additional discussion of mediators has also been added to the text (Line 421-422 and 427-428).

Conclusions.

Pg 12 Ln 410: Suggest changing ‘its growth physiology’ to ‘their growth physiology’

This has been changed

Reviewer 2 Report

In the present work, the authors have made an interesting description of the phenotypic isolation and genetic characterization of two strains belonging to the Idiomarina genus (SN11 and FeN1) which presents an interesting and still little explored electrochemical activity.

Thanks to the effort in phenotypic characterization and biological assay, the authors postulated the hypothesis that both strains are obligate heterotrophic, dependent exclusively on the absorption and degradation of amino acids as the primary source of carbon for growth.

Despite this, thanks to the isolation effort and cathode enrichment testing, and the ability of strains to sustain electron absorption over time in an electrochemical system, the authors were able to postulate a possible role of EET exclusively for respiration.

The genomics investigation of both strains highlighted new candidate genomic pathways involved in EET has been characterized, although more in-depth investigations are needed to better characterize the role of these genes in the EET process.

The article highlighted new as yet unexplored metabolic potentials in which microbes can directly utilize the flow of electrons from both biotic and abiotic origin as an energy supplement for growth, implementing our knowledge of microbial-mediated processes in both natural and artificial environments.

Author Response

We thank the reviewer for their comments and feed back. Though no major changes were requested we endeavored to enhance that analysis or iron metabolisms by Idiomarina using the FeGenie software tool. This analysis has been added to the methods (Line 146-150) and the results and discussion (Line 346-353). Additionally we have made all typographical and clarifying suggestion of reviewer 1. Thanks again for your time and consideration.